# Superconductivity from the condensation of topological defects in a quantum spin-Hall insulator

Yuhai Liu[1], Zhenjiu Wang[2], Toshihiro Sato[2], Martin Hohenadler[2], Chong Wang[3], Wenan Guo [1,4] &
Fakher F. Assaad[2]

The discovery of quantum spin-Hall (QSH) insulators has brought topology to the forefront of condensed matter physics. While a QSH state from spin-orbit coupling can be fully understood in terms of band theory, fascinating many-body effects are expected if it instead results from spontaneous symmetry breaking. Here, we introduce a model of interacting Dirac fermions where a QSH state is dynamically generated. Our tuning parameter further allows us to destabilize the QSH state in favour of a superconducting state through proliferation of charge-2e topological defects. This route to superconductivity put forward by Grover and Senthil is an instance of a deconfined quantum critical point (DQCP). Our model offers the possibility to study DQCPs without a second length scale associated with the reduced symmetry between field theory and lattice realization and, by construction, is amenable to large-scale fermion quantum Monte Carlo simulations.

[1] Department of Physics, Beijing Normal University, Beijing 100875, China. [2] Institut für Theoretische Physik und Astrophysik, Universität Würzburg, Am Hubland, 97074 Würzburg, Germany. [3] Perimeter Institute for Theoretical Physics, Waterloo, ON N2L 2Y5, Canada. [4] Beijing Computational Science Research Center, Beijing 100193, China. Correspondence and requests for materials should be addressed to W.G. (email: waguo@bnu.edu.cn) or to F.F.A. (email: fakher.assaad@physik.uni-wuerzburg.de)

In the Kane-Mele model for the quantum spin-Hall (QSH) insulator[1], the original SU(2) spin symmetry is explicitly broken by spin-orbit coupling. Here, we instead consider the case where this symmetry is preserved by the Hamiltonian but spontaneously broken by an interaction-generated QSH state[2]. At the mean-field level, the latter is characterised by an SO(3) order parameter constant in space and time and a band structure with a non-trivial $\mathbb{Z}_2$ topological index[1,3,4]. Long-wavelength fluctuations of this order parameter include in particular the Goldstone modes that play a key role for phase transitions to, e.g., a Dirac semimetal. Such a transition, illustrated in Fig. 1a, is described by a Gross-Neveu-Yukawa field theory[5,6] with QSH order encoded in a mass in the underlying Dirac equation. Fluctuations can also take the form of topological ('skyrmion') defects that correspond to a non-trivial winding of the order parameter vector. Due to the topological band structure of the QSH state, such skyrmions carry electric charge 2e[7]: as illustrated in the Supplementary Discussion, the insertion of a skyrmion in a system with open boundaries pumps a pair of charges from the valence to the conduction band through the helical edge states. The condensation of skyrmion defects—which coincides with the destruction of the QSH state—represents a route to generate a superconducting (SC) state.

A direct QSH-SC phase transition (Fig. 1a) is an instance of a deconfined quantum critical point (DQCP)[8–10], the concept of which relies on the topological defects of one phase carrying the charge of the other phase. Defect condensation then provides a mechanism for a continuous transition between two states with different broken symmetries (SO(3) for QSH, U(1) for SC) that is forbidden by Landau theory. Despite considerable numerical efforts[11,12], DQCPs remain a subject of intense debate. Important questions include their very nature—weakly first order or continuous[10]—and the role of emergent symmetries[13]. One of the difficulties lies in the fact that previous lattice realisations[12,14,15] involve antiferromagnetic (AFM) and valence bond solid (VBS) phases. For the widely studied square lattice, the VBS state breaks the discrete $\mathbb{Z}_4$ rotation symmetry, whereas the field theory has a U(1) symmetry. The latter is recovered on the lattice exactly at the critical point, but in general the $\mathbb{Z}_4$ symmetry breaking term is relevant. The additional length scale at which the $\mathbb{Z}_4$ symmetry becomes visible obscures the numerical analysis. In the field theory, this translates into the notion that quadruple skyrmion addition (monopole) events of the AFM SO(3) order parameter are irrelevant at criticality but proliferate slightly away from this point to generate the VBS state[8,16,17]. Hence, the theory is subject to a dangerously irrelevant operator. This complication is completely avoided in the model introduced here, where the DQCP separates QSH and SC rather than AFM and VBS phases. QSH and AFM order are both described by an SO(3) order parameter. However, instead of the $\mathbb{Z}_4$ symmetry broken by the lattice VBS state, the SC phase breaks the same global U(1) gauge symmetry (charge conservation) on the lattice and in the continuum.

Therefore, the number of fat skyrmion defects[7] with charge 2e is conserved and monopoles are absent.

The exciting prospects of (i) SC order from topological defects of a spontaneously generated QSH state and (ii) a monopole-free realisation of a DQCP motivate the search for a suitable lattice model amenable to quantum Monte Carlo simulations without a sign problem. Such efforts are part of the recent surge of designer Hamiltonians aimed at studying exotic phases and phase transitions[18–24]. In this article we introduce and solve a model that realises the quantum phase transition between QSH and SC states.

## Results

**Model.** Our starting point is a tight-binding model of Dirac fermions in the form of electrons on the honeycomb lattice with nearest-neighbour hopping (see Fig. 1b), as described by the Hamiltonian

$$\hat{H}_t = -t \sum_{\langle i,j \rangle} (\hat{c}_i^\dagger \hat{c}_j + \text{H.c.}). \tag{1}$$

The spinor $\hat{c}_i^\dagger = (\hat{c}_{i,\uparrow}^\dagger, \hat{c}_{i,\downarrow}^\dagger)$, where $\hat{c}_{i,\sigma}^\dagger$ creates an electron at lattice site $i$ with spin $\sigma$. Equation (1) yields the familiar graphene band structure with gapless, linear excitations at the Dirac points[25]. A suitable interaction that generates the above physics is

$$\hat{H}_\lambda = -\lambda \sum_{\hexagon} \left( \sum_{\langle\langle i,j \rangle\rangle \in \hexagon} \mathrm{i}\nu_{ij} \hat{c}_i^\dagger \boldsymbol{\sigma} \hat{c}_j + \text{H.c.} \right)^2. \tag{2}$$

The first sum is over all the hexagons of a honeycomb lattice with $L \times L$ units cells and periodic boundary conditions. The second sum is over all pairs of next-nearest-neighbour sites of a hexagon, see Fig. 1b. The quantity $\nu_{ij} = \pm 1$ is identical to the Kane-Mele model[1]; for a path from site $i$ to site $j$ (connected by $\boldsymbol{R}_{ij}$, see Fig. 1b) via site $k$, $\nu_{ij} = \hat{\boldsymbol{e}}_z \cdot (\boldsymbol{R}_{ik} \times \boldsymbol{R}_{kj})/|\hat{\boldsymbol{e}}_z \cdot (\boldsymbol{R}_{ik} \times \boldsymbol{R}_{kj})|$ with $\hat{\boldsymbol{e}}_z$ a unit vector perpendicular to the honeycomb plane. Finally, $\boldsymbol{\sigma} = (\sigma^x, \sigma^y, \sigma^z)$ with the Pauli spin matrices $\sigma^\alpha$.

The rationale for this choice of interaction is easy to understand. Without the square, and taking just one of the three Pauli matrices, Eq. (2) reduces to the Kane-Mele spin-orbit coupling that explicitly breaks the SO(3) spin symmetry. In contrast, the latter is preserved by $\hat{H}_\lambda$ but spontaneously broken by long-range QSH order. For $\lambda > 0$, the model defined by $\hat{H} = \hat{H}_t + \hat{H}_\lambda$ can be simulated without a sign problem by auxiliary-field quantum Monte Carlo methods[26–28]. In the following, we set $t = 1$ and consider a half-filled band with one electron per site.

A mean-field decomposition of Eq. (2) with order parameter field $\boldsymbol{N}_{\hexagon} = \left\langle \sum_{\langle\langle i,j \rangle\rangle \in \hexagon} \mathrm{i}\nu_{ij} \hat{c}_i^\dagger \boldsymbol{\sigma} \hat{c}_j + \text{H.c.} \right\rangle$ suggests a transition from the Dirac semimetal to a QSH state at a critical value $\lambda_{c1} > 0$. However, it is highly non-trivial if the associated saddle point is stable. In fact, s-wave pair hopping processes arise upon expanding the square in Eq. (2) and can lead to super-conductivity[29]. The exact phase diagram can be obtained by quantum Monte Carlo simulations. Remarkably, as illustrated in Fig. 1a, we find two distinct phase transitions. First, from the semimetal to a QSH state at $\lambda_{c1}$, then from the QSH state to an s-wave SC at $\lambda_{c2} > \lambda_{c1}$.

**Order parameters.** The semimetal-QSH transition involves the breaking of spin rotation symmetry and is expected to be in the O(3) Gross-Neveu universality class for $N = 8$ Dirac fermions (two sublattices, two Dirac points, $\sigma = \uparrow, \downarrow$). The local vector order parameter takes the form of a spin-orbit coupling,

$$\hat{\boldsymbol{O}}_{r,\delta}^{\text{QSH}} = \mathrm{i}\hat{c}_r^\dagger \boldsymbol{\sigma} \hat{c}_{r+\delta} + \text{H.c.}, \tag{3}$$

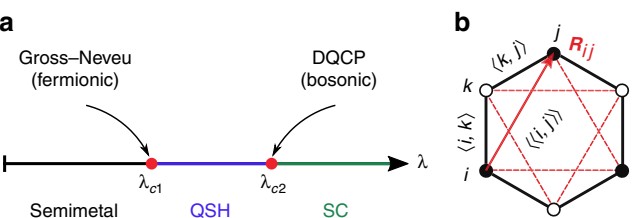

**Fig. 1** Phase diagram and model. **a** Schematic ground-state phase diagram with semimetallic, QSH, and SC phases. **b** Illustration of nearest- and next-nearest neighbours and the vector $\boldsymbol{R}_{ij}$ on a honeycomb lattice plaquette

where $\boldsymbol{r}$ corresponds to a unit cell labelling a hexagon, and $\boldsymbol{r} + \boldsymbol{\delta}$ runs over all next-nearest neighbours. Because this order parameter is a lattice regularisation of the three QSH mass terms in the Dirac equation, long-range order implies a mass gap[1]. To study the phase transition, we computed the associated susceptibility

$$\chi^O_{\boldsymbol{\delta},\boldsymbol{\delta'}}(\boldsymbol{q}) = \frac{1}{L^2} \sum_{\boldsymbol{r},\boldsymbol{r'}} \int_0^\beta d\tau e^{i\boldsymbol{q}\cdot(\boldsymbol{r}-\boldsymbol{r'})} \langle \hat{O}_{\boldsymbol{r},\boldsymbol{\delta}}(\tau)\hat{O}_{\boldsymbol{r'},\boldsymbol{\delta'}}(0)\rangle. \quad (4)$$

Here, $\langle \hat{O}_{\boldsymbol{r},\boldsymbol{\delta}}(\tau)\rangle = 0$ by symmetry for finite $L$ and we concentrate on the largest eigenvalue of $\chi^O_{\boldsymbol{\delta},\boldsymbol{\delta'}}(\boldsymbol{q})$ (see Supplementary Notes), henceforth denoted as $\chi^O(\boldsymbol{q})$. To detect the transition, we consider the renormalisation-group invariant correlation ratio

$$1 - \frac{\chi^O(\boldsymbol{Q}+\Delta\boldsymbol{q})}{\chi^O(\boldsymbol{Q})} = R^O_\chi\left(L^{1/\nu}(\lambda-\lambda^O_c), L^{-\omega}\right) \quad (5)$$

with $|\Delta\boldsymbol{q}| = \frac{4\pi}{\sqrt{3}L}$, the ordering wavevector $\boldsymbol{Q} = 0$, the correlation length exponent $\nu$ and the leading corrections-to-scaling exponent $\omega$. We set the inverse temperature $\beta = L$ in our simulations based on the assumption of a dynamical critical exponent $z = 1$[30]. In contrast to previous analyses of Gross-Neveu criticality[31,32] we use susceptibilities rather than equal-time correlators to suppresses background contributions to the critical fluctuations.

**Numerical results**. The results for the semimetal-QSH transition are shown in Fig. 2. The finite-size estimate of the critical value, $\lambda^{QSH}_{c1}(L)$, corresponds to the crossing point of $R^{QSH}_\chi$ for $L$ and $L+6$. Extrapolation to the thermodynamic limit (inset of Fig. 2a) yields $\lambda^{QSH}_{c1} = 0.0187(2)$. As shown in the Supplementary Fig. 4, the single-particle gap is nonzero for $\lambda > \lambda^{QSH}_{c1}$. The correlation length

exponent was estimated from[11]

$$\frac{1}{\nu^O(L)} = \frac{1}{\log r}\log\left(\frac{\frac{d}{d\lambda}R^O_\chi(\lambda, rL)}{\frac{d}{d\lambda}R^O_\chi(\lambda, L)}\right)\Bigg|_{\lambda=\lambda^O_c(L)} \quad (6)$$

with $r = \frac{L+6}{L}$. A similar equation can be used to determine the exponent $\eta$ from the divergence of the susceptibility ($\chi^O \propto L^{2-\eta}$) at criticality (see SI). Aside from a polynomial interpolation of the data as a function of $\lambda$ for each $L$, this analysis does not require any further fitting and, by definition, converges to the correct exponents in the thermodynamic limit with rate $L^{-\omega}$. While existing estimates of the critical exponents vary[31–33], the values $1/\nu = 1.14(9)$ and $\eta = 0.79(5)$ from Fig. 2 are consistent with $\nu = 1.02(1)$ and $\eta = 0.76(2)$ from previous work[32]. This suggest that the semimetal-QSH transition is in the same universality class as the semimetal-AFM transition[31,32,34].

To detect SC order, we used the order parameter

$$\hat{O}^{SC}_{\boldsymbol{r},\tilde{\boldsymbol{\delta}}} = \frac{1}{2}\left(\hat{c}^\dagger_{\boldsymbol{r}+\tilde{\boldsymbol{\delta}},\uparrow}\hat{c}^\dagger_{\boldsymbol{r}+\tilde{\boldsymbol{\delta}},\downarrow} + \text{H.c.}\right) \quad (7)$$

where $\boldsymbol{r} + \tilde{\boldsymbol{\delta}}$ runs over the two orbitals of unit cell $\boldsymbol{r}$. As before, we computed the corresponding susceptibility and used $\beta = L$ in anticipation of $z = 1$. Figure 3 shows that, within the very small error bars, the critical value for SC order $\lambda^{SC}_{c2} = 0.0332(2)$ and the critical value for the disappearance of long-range QSH order $\lambda^{QSH}_{c2} = 0.03322(3)$ are identical, suggesting a direct QSH-SC transition. At this transition, the single-particle gap remains of order one and we find no evidence for a first-order transition for the available system sizes (See Supplementary Fig. 4).

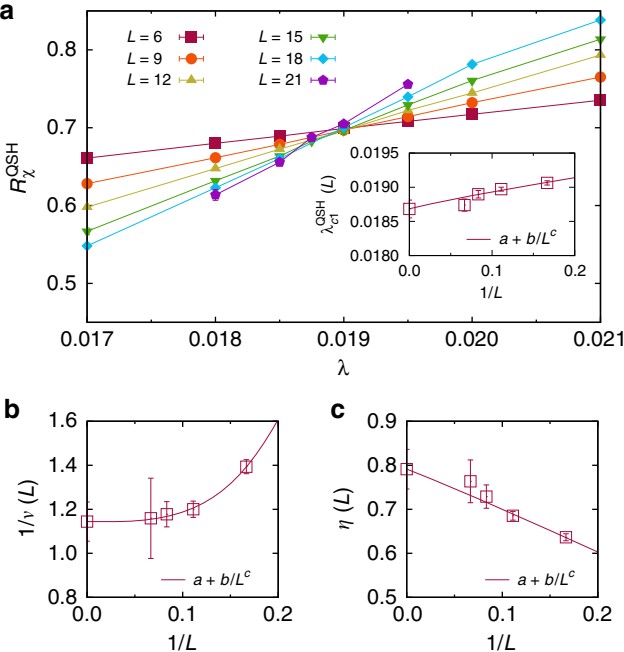

**Fig. 2** Gross-Neveu semimetal-QSH transition. **a** Correlation ratio $R^{QSH}_\chi$ [Eq. (5)] for different system sizes $L$. The extrapolation of the crossing points of $R^{QSH}_\chi$ for $L$ and $L+6$ in the inset gives the critical value $\lambda_{c1} = 0.0187(2)$. **b** Finite-size scaling based on Eq. (6) gives an inverse correlation length exponent $1/\nu = 1.14(9)$. **c** Estimation of the anomalous dimension $\eta = 0.79(5)$

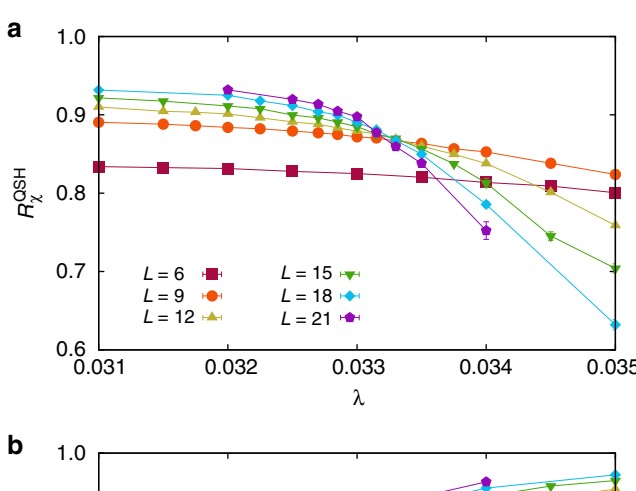

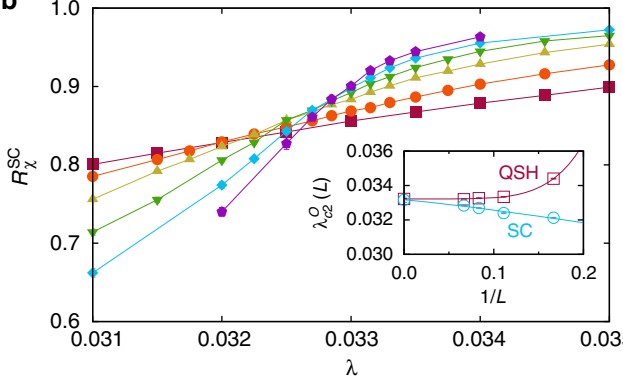

**Fig. 3** Deconfined QSH-SC transition. **a** Correlation ratio $R^{QSH}_\chi$ and **b** correlation ratio $R^{SC}_\chi$ for different system sizes $L$. The extrapolation of the crossing points for $L$ and $L+6$ using the form $a + b/L^c$ (see inset of **b**) gives $\lambda^{QSH}_{c2} = 0.03322(3)$ and $\lambda^{SC}_{c2} = 0.0332(2)$

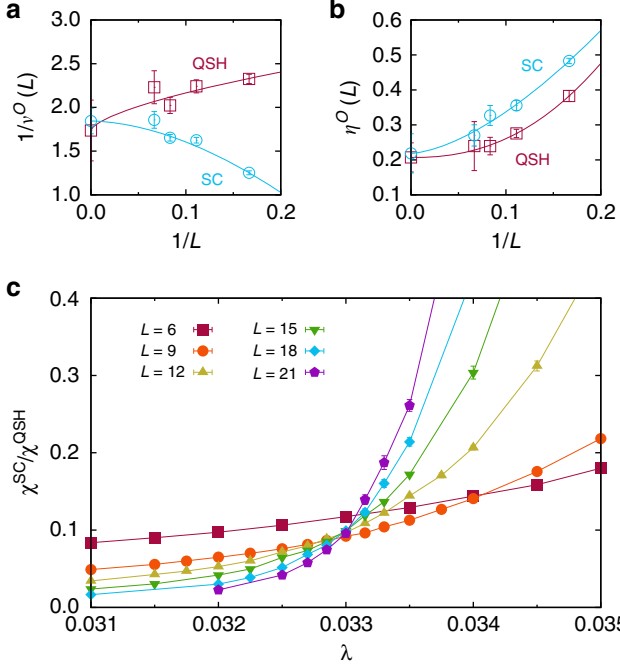

**Fig. 4** Critical exponents for the QSH-SC transition. **a, b** Critical exponents $1/\nu^{SC} = 1.8(2)$, $1/\nu^{QSH} = 1.7(4)$, $\eta^{SC} = 0.22(6)$ and $\eta^{QSH} = 0.21(5)$ from finite-size scaling of the crossing points for $L$ and $L + 6$. **c** Ratio of the QSH and SC susceptibilities for different system sizes $L$

The observed s-wave symmetry of the SC state emerges directly from the perspective of Dirac mass terms. In $2 + 1$ dimensions and for $N = 8$ Dirac fermions, there exist numerous quintuplets of anti-commuting mass terms that combine different order parameters in a higher SO(5) symmetry group[35]. A well-known example relevant for DQCPs are the three AFM and two VBS mass terms. Here, the three QSH mass terms form a quintuplet with the two s-wave SC mass terms. The resulting SO(5) order parameter allows for a very natural derivation of the Wess-Zumino-Witten term[36,37], crucial for the DQCP, by integrating out the (massive) Dirac fermions[38].

As argued in the introduction, the QSH-SC problem is free of monopoles, so that our lattice model represents an improved model to study the DQCP. Although simulations for fermions are limited to smaller system sizes than for bosons, severe size effects due to monopoles[11] can be expected to be absent. Figure 4 shows a finite-size analysis for the correlation length exponent and the anomalous dimension, based on either the QSH or the SC correlation ratio. The resulting estimates $\eta^{QSH} = 0.21(5)$ and $\eta^{SC} = 0.22(6)$ are compatible with those from loop models[12] where $\eta^{AFM} = 0.259(6)$ and $\eta^{VBS} = 0.25(3)$. An alternative analysis described in the Supplementary Methods yields similar values. Given the very similar anomalous dimensions $\eta^{QSH}$ and $\eta^{SC}$ of QSH and SC fluctuations, the ratio of the QSH and SC susceptibilities is expected to be a renormalisation group invariant, as confirmed by Fig. 4c. However, a crossing of different curves at $\lambda_{c2}$ is a necessary but not a sufficient condition for an emergent SO(5) symmetry at the DQCP. In fact, a continuous transition with emergent SO(5) symmetry can be essentially excluded here in the light of the condition $\eta > 0.52$ from the conformal bootstrap method[39]. The latter also yields a bound of $1/\nu < 1.957$ for a unitary conformal field theory with only one tuning parameter[40] that is satisfied by $1/\nu^{SC} = 1.8(2)$ and $1/\nu^{QSH} = 1.7(4)$ from Fig. 4a but not by the value $1/\nu = 2.24(4)$ reported before[11]. Simulations of the monopole-free model on even larger lattices are required for a conclusive answer.

## Discussion

Our model provides a realisation of a QSH insulator emerging from spontaneous symmetry breaking. The corresponding SO(3) order parameter permits both long-wavelength Goldstone modes and topological skyrmion defects. By means of a single parameter $\lambda$, we can trigger continuous quantum phase transitions to either a semimetal or an s-wave SC state. For the semimetal-QSH transition, the critical exponents are consistent with Gross-Neveu universality[31,32]. The QSH-SC transition is of particular interest since it provides a monopole-free, improved model of deconfined quantum criticality with only one length scale. The mechanism for SC order from the QSH state is the condensation of skyrmion defects of the QSH order parameter with charge 2e. For the QSH-SC transition, our values of the anomalous dimension match those of previous work on the AFM-VBS transition[12,15], which are inconsistent with results from conformal bootstrap studies if an SO(5) symmetry emerges at the critical point (as supported by numerical and analytical studies). One possible resolution is the scenario of 'pseudo-criticality' where the fixed point lies slightly outside the accessible parameter space and the RG flow becomes very slow[10,12,41,42]. In contrast, our estimate of $1/\nu$ is still within the conformal bootstrap bound[40], although a bound-violating result is not completely ruled out given the numerical uncertainty. Consequently, it is of considerable interest to exploit the full potential of quantum Monte Carlo methods in order to access even larger lattices. Other promising approaches that can shed further light on DQCPs make use of a lattice discretisation scheme based on projection onto a Landau level, so that models with explicit SO(5) symmetry can be considered[43].

In traditional realisations of deconfined criticality in spin models, the finite-size analysis is subtle due to the dangerously irrelevant perturbation (the monopoles)[11]. The absence of the latter is a major advantage of the fermionic model studied here and makes the interpretation of the finite-size scaling relatively straightforward. A monopole-free realisation of DQCPs is impossible in traditional spin models because of a quantum anomaly[10] for the SO(3) × U(1) symmetry in the effective field theory. Essentially, this anomaly rules out any (local) lattice realisation of DQCP with exact SO(3) × U(1) symmetry. In the standard setting, what is being realised on the lattice is SO(3) × $\mathbb{Z}_4$ ($\mathbb{Z}_4$ being the lattice $C_4$), and the full SO(3) × U(1) is emergent only in the infrared limit, leaving the U(1) → $\mathbb{Z}_4$ anisotropy as a dangerously irrelevant perturbation. (In fact even the SO(3) × $\mathbb{Z}_4$ is still anomalous. This anomaly is matched by the non-onsite nature of lattice rotation symmetries[44]). In contrast, the model in our work has the exact U(1) symmetry (charge conservation). In terms of anomalies, this is possible because of the existence of microscopic degrees of freedom (the fermions) that carry 'fractional' symmetry quantum numbers (half-spin and half-charge in terms of Cooper pair charges). In a more formal language, the anomaly is eliminated by properly extending the global symmetry (hence allowing smaller representations such as spin-1/2). An even simpler extension of the symmetry that eliminates the anomaly is SU(2) × U(1), meaning that microscopically there are charged spinless bosons, together with both charged and neutral spin-1/2 bosons. A challenge for future studies is to find a reasonably simple Hamiltonian that realises a DQCP and is amenable to sign-free bosonic QMC simulations in, e.g., the stochastic series expansion representation[45].

The SC phase generated from skyrmion defects motivates further investigations. Its vortex excitations carry a spin-1/2 degree of freedom[7], so that in the quantum critical fan thermal melting produces a gas of charged spinons[46]. It is also possible to add an independent attractive Hubbard interaction to explore a semimetal-QSH-SC tricritical point (as opposed to the recently discovered semimetal-AFM-VBS tricritical point[24]) with

predicted SO(5) Gross-Neveu criticality[47,48]. The vector form of $\hat{H}_\lambda$ makes it straight forward to reduce the SO(3) QSH symmetry to U(1) and thereby investigate an easy-plane realisation of DQCPs with a U(1) × U(1) symmetry on the lattice. Work along these directions is in progress.

## Methods

**Quantum Monte Carlo.** We employed the ALF[49] implementation of the auxiliary-field finite-temperature quantum Monte Carlo method[26–28]. The interaction term is written as a perfect square with negative prefactor ($\lambda > 0$), allowing for a decomposition in terms of a real Hubbard-Stratonovitch field. For each field configuration, time-reversal symmetry holds and the eigenvalues of the fermion matrix occur in complex conjugate pairs[50–52]. At low temperatures, the scales of the imaginary-time propagation do not fit into double precision real numbers and we have used methods to circumvent this issue [53]. The imaginary-time discretisation was $\Delta\tau = 0.2$. For reasons explained in the Supplementary Notes, we chose a symmetric Trotter decomposition that minimises discretization errors. Reported errors and error bars in figures correspond to standard errors.

## Data availability

The datasets generated during the current study are available from the corresponding author on reasonable request.

## Code availability

ALF[49] is open-source software that is available from the corresponding author on request.

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

## Acknowledgements
We acknowledge illuminating discussions with T. Grover, and with J. Hofmann regarding extensions of the ALF package[49]. We thank the Gauss Centre for Super-computing (SuperMUC at the Leibniz Supercomputing Centre) for generous allocation of supercomputing resources  and acknowledge financial support from the Deutsche Forschungsgemeinschaft (DFG) grant AS120/14-1 for further development of the ALF package. TS thanks the DFG for financial support from grant number AS120/15-1. M.H. and Z.W. are supported by the DFG collaborative research centre SFB1170 ToCoTronics (project C01), Y.L. and W.G. by NSFC under grants nos. 11775021 and 11734002. Research at Perimeter Institute (C.W.) is supported by the Government of Canada through the Department of Innovation, Science and Economic Development Canada and by the Province of Ontario through the Ministry of Research, Innovation and Science.

## Author contributions
C.W., W.G. and F.F.A. conceived the project. Y.L. and Z.W. implemented the model in the ALF package and carried out the simulations under the guidance of T.S. and M.H. All authors participated in the interpretation of the results and the writing of the manuscript.

## Additional information

**Competing interests:** The authors declare no competing interests.

