## [Peer Review File · Nature Communications]

Reviewers' comments:

Reviewer #1 (Remarks to the Author):

The authors present a careful quantum Monte Carlo study of a model of interacting fermions on the 2D honeycomb lattice, and show that this model exhibits two quantum phase transitions, one from a Dirac semimetal with 4 two-component Dirac cones to an interaction-induced quantum spin Hall insulator (e.g., a topological Mott insulator), and a subsequent one from the interaction-induced QSH insulator to a spin-singlet, s-wave superconductor. This second transition is the main focus of the study.

In short, the authors have found and studied a model that realizes the scenario proposed by Grover and Senthil in 2009 (Ref. 7) according to which the condensation of skyrmions in the QSH order parameter, which carry charge $2e$ because of the spin Hall effect in the QSH phase, produces a direct, continuous quantum phase transition to a superconducting phase. This is a very nice instance of a deconfined quantum critical point (DQCP), an exotic type of Landau-forbidden phase transition originally proposed in Ref. 8-9. As the authors point out, their model has a special property that makes the numerical observation of a DQCP more feasible as compared to that of the Néel-VBS DQCP for spin models on the square lattice.

In my opinion this is an important advance pertaining to the fields of strongly correlated systems, quantum criticality, and topological materials, that deserves wide dissemination. I support publication.

However, I am confused by the sentence "In fact, a continuous transition with emergent $SO(5)$ symmetry can be essentially excluded here in the light of the condition $\eta > 0.52$ from the conformal bootstrap method." If I understand correctly, the authors' main claim is that the QSH-SC QCP they have found is a realization of a DQCP with emergent $SO(5)$ symmetry (as argued, e.g., in the paragraph just below Fig. 3). The sentence above seems to precisely revoke that claim. I think the authors really mean that at the values of L and β considered their numerically found exponents violate the conformal bootstrap bounds, and that studies at larger L and β are required to reduce the uncertainty in the QMC exponents and truly determine whether one has a DQCP with $SO(5)$ symmetry or not. Either way, this sentence should be clarified.

Also, it may be helpful for the non-expert reader to expand a bit on what the "pseudo-criticality" and "walking coupling constant" scenarios are.

Reviewer #2 (Remarks to the Author):

This paper starts from the two-band Hamiltonian describing non-interacting graphene at half filling.

The goal of this paper is to study the phase diagram of this Hamiltonian, when perturbed by a certain local attractive quartic interaction that is supported on the sites of the hexagonal cell of graphene with the dimensionfull coupling $\lambda \geq 0$.

It is argued that there are two quantum critical points $\lambda_{c1} > 0$ and

$\lambda_{c2} > \lambda_{c1} > 0$
separating three phases:

1) For sufficiently small $\lambda \geq 0$, $0 \leq \lambda < \lambda_{c1}$,
the ground state is semi-metallic, i.e.,
the spectrum is gapless with a vanishing density of states
at the Fermi energy associated to two inequivalent Dirac points.

2) For intermediary $\lambda_{c1} < \lambda < \lambda_{c2}$,
the ground state is gapped with a quantized value for the quantum spin Hall
conductivity.

3) For large $\lambda_{c2} < \lambda$,
the ground state is gapped with superconducting long-range order
in the symmetry class CI.

The main result of this paper is to establish the existence of the two quantum
critical points λ_{c1} and λ_{c2} . The quantum critical point
 λ_{c1} belongs to the fermionic Gross-Neveu universality class
in (2+1)-dimensional spacetime. The quantum critical point
 λ_{c2} is a deconfined quantum critical point (DQCP)
in (2+1)-dimensional spacetime
separating a gapped phase that breaks an $SU(2)$
spin symmetry spontaneously down to a $U(1)$ subgroup
from a gapped phase that breaks the $U(1)$ symmetry for electric charge
conservation down to a discrete subgroup \mathbb{Z}_2 .

The first quantum critical point λ_{c1} has already been studied
within a mean-field approximation about ten years ago.
To the best of my knowledge, this is the first lattice model
that realizes the second quantum critical point λ_{c2} .

The method used in this paper to identify the three ground states
and the two quantum-critical point separating them is
auxiliary-field finite-temperature quantum Monte Carlo method.
Although I am not an expert in computational physics, I found
convincing the finite-size scaling analysis done in support of
the phase diagram.

However, the contextualization was confusing:

1) The abstract and first paragraph do not meet the quality required for
Nature Communication. They read like a laundry list of concepts without
a binding sauce. They lack focus, conciseness, and a punch line.
The claim '... the SC order may be
understood as emerging from a gapless spin liquid normal state.'
is not supported by original calculations in the paper. Two lines are
devoted to this claim by which the reader is referred to Refs. 7 and 44
for more detailed explanation. What is '... an improved model ...'
in the last line of the abstract?

2) The claim 'Due to the topological nature of the QSH state itself,
such skyrmions carry electric charge $2e$ '
is ambiguous because the authors do not define precisely what is meant
by topology. I believe that the spin quantum Hall effect is a consequence

of the identity $\pi_0(\text{Target space}) = \mathbb{Z}$, while the existence of Skyrmions is a consequence of the identity $\pi_2(S^2) = \mathbb{Z}$. There are several papers prior to Ref 7 that imply the charge assignment $2e$ when solving the Dirac equation in two-dimensional space (or any lattice model regularizing the Dirac equation) subjected to a Skyrmion background.

I do not understand the 'In contrast to a weakly coupled Bardeen-Cooper-Schrieffer-type SC, its vortices enclose a spin-1/2 degree of freedom corresponding to the fractionalized QSH order parameter.' The canonical BCS superconducting state describes a system with spin-rotation symmetry. Vortices support normal-state excitations, namely electrons. Evidently, electrons carry a spin-1/2, which is a good quantum number. I have no idea what is meant by a 'fractionalized QSH order parameter' in the SC phase for which $SU(2)$ spin symmetry holds. It should be possible to repeat the ideas from Ref 7 in a better way.

3) I would also have given a reference to a famous paper by Haldane in addition to Ref 8 in:

'... but proliferate slightly away from this point to generate the VBS state.' The discussion of Skyrmion-number conservation and hedgehogs confuses me. Reference 7 states that Skyrmion number conservation only applies to fat Skyrmions.

4) I found the sentence

'However, it is highly non-trivial if the associated saddle point is stable.' poorly written. There are many ways to decouple a quartic interaction (there are many instabilities by which distinct quadratic forms acquire an expectation value), i.e., there are competing instabilities.

5) I was confused by

'Here, we omitted the vanishing background term ...'

and

'In contrast to previous analyses of Gross-Neveu criticality 29,30 we use susceptibilities rather than equal-time correlators to suppress background contributions to the critical fluctuations.'

If spontaneous symmetry breaking takes place, one should replace moments by cumulants. Are the authors referring to the C number arising from long-range order at the ordering wave vector?

6) I could not appreciate the claim containing

'... compare favourably ...'.

7) The discussion around the sentence 'One possible resolution is the scenario of 'pseudo-criticality' or 'walking coupling constant'.' was too short for me to be of any use.

8) The discussion around the sentence

'Other promising approaches that can shed further light on DQCPs make use of a lattice discretisation scheme based on projection onto a Landau level that does not break continuum symmetries.' was too short for me to be of any use.

9) I could not follow the discussion in the paragraph
``A monopole-free realisation of DQCPs is impossible in
traditional settings ...'.

I am hesitant to recommend this paper for publication in
Nature Communications, assuming that items 1-9) have been fixed.

This is so because I find that this paper

a) is less creative than Ref. 22, a paper sharing many co-authors,
while the methodology is the same,

b) is too close relative to Refs. 19 and 22,

c) and adds nothing conceptually relative to Ref. 7.

The justification for publication in Nature Communications,

namely the discovery of a lattice model that realizes

the DQCP proposed in Ref. 7 within the limitations

of auxiliary-field finite-temperature quantum Monte Carlo,

is on the cusp of marginality. It seems that the authors believe

that the conservation of the Skyrmion number justifies

publication in Nature Communications. However, there is no

original discussion of this property in this paper and

the numerics done in this paper do not provide new

insights on this property (except for the indirect consequence

that numerics are ``cleaner').

It would thus not be a crime to publish

or not publish this paper in Nature Communications,

assuming that items 1-9) have been fixed.

Reply to Reviewer # 1

Thank you for the time you have taken to read our manuscript. The comments,

However, I am confused by the sentence “In fact, a continuous transition with emergent SO(5) symmetry can be essentially excluded here in the light of the condition $\eta > 0.52$ from the conformal bootstrap method.” If I understand correctly, the authors’ main claim is that the QSH-SC QCP they have found is a realization of a DQCP with emergent SO(5) symmetry (as argued, e.g., in the paragraph just below Fig. 3). The sentence above seems to precisely revoke that claim. I think the authors really mean that at the values of L and β considered their numerically found exponents violate the conformal bootstrap bounds, and that studies at larger L and β are required to reduce the uncertainty in the QMC exponents and truly determine whether one has a DQCP with SO(5) symmetry or not. Either way, this sentence should be clarified. Also, it may be helpful for the non-expert reader to expand a bit on what the “pseudo-criticality” and “walking coupling constant” scenarios are.

are all related to the interpretation of our results in light of the conformal bootstrap bounds for an emergent SO(5) symmetry.

Our results support the following scenarios:

- 1) The theory of deconfined quantum criticality (DQC) does not necessarily require an emergent SO(5) symmetry. One can hence assume a unitary CFT in $2 + 1$ dimensions akin to the DQC transition that does not support emergent SO(5) symmetry. In this case, larger system sizes would be necessary to confirm the absence of an emergent SO(5) symmetry. Note that here the conformal bootstrap bound $1/\nu < 1.957$ still has to hold. This is presently supported by our data, but our error bars are still rather large. More work is required to clarify this point. In contrast, the results of Ref. [1] contradict this bound.
- 2) The numerical simulations of Refs. [2, 3] show compelling evidence for emergent SO(5) symmetry and drifting exponents. To explain this anomalous behaviour, one can for example postulate the existence of a unitary CFT in say $2 + \epsilon + 1$ dimensions ($\epsilon > 0$), akin to the DQC transition and supporting an emergent SO(5) symmetry. If this is the case, the conformal bootstrap bounds cited in our paper no longer apply to our numerical simulations. If ϵ is *small*, the RG flow will become very slow in the vicinity of the critical point. The fixed point will never be reached since our simulations are restricted to two spatial dimensions, but the correlation length required to resolve this behaviour exceeds the achievable lattice sizes. These ideas underlie the notion of pseudo-criticality [4, 5].

We have revised the manuscript to make this point clearer.

Reply to Reviewer # 2

Thank you for the detailed assessment of our manuscript. We agree with your summary of the major achievements of our paper. The quantum spin Hall state and the underlying spin-orbit coupling are dynamically generated. The associated $SO(3)$ spin symmetry is thus broken down to $U(1)$ so that, as you correctly mention, the state is characterized by a quantized spin Hall conductivity. Note that breaking down the $U(1)$ symmetry further to Z_2 would lift the quantization.

We first comment point by point on the questions you raise and then expand on why we do not agree with the arguments based on which you hesitate to recommend publication in Nature Communications.

1) The abstract and first paragraph do not meet the quality required for Nature Communication. They read like a laundry list of concepts without a binding sauce. They lack focus, conciseness, and a punch line. The claim "... the SC order may be understood as emerging from a gapless spin liquid normal state." is not supported by original calculations in the paper. Two lines are devoted to this claim by which the reader is referred to Refs. 7 and 44 for more detailed explanation. What is "... an improved model ..." in the last line of the abstract?

We agree that the abstract did not meet the style requirements of the journal and have significantly rewritten it. In particular, we have tried to avoid any jargon and point out the highlights of our results in simple words. We have also omitted the references to spin liquids in the abstract and the introduction since they are not essential for the narrative.

In statistical mechanics, an improved model is one where corrections to scaling are suppressed. As a specific example, one can mention the Blume-Capel model for the Ising universality class (see Ref. [6] and references therein). In the theory of DQCPs, vortices of the superconducting order parameter carry a spin-1/2 object with unit electric charge as well as a $U(1)$ gauge charge. (In contrast, electrons carry no gauge charge. In fact, in the vicinity of the QSH-SC transition and up to energy scales set by the hopping matrix element, the single-particle excitations are gapped.) The deconfined state of these spin-1/2 objects would correspond to a spin liquid.

2) In the sentence **Due to the topological nature of the QSH state itself, such skyrmions carry electric charge $2e$, the topological nature of the QSH state itself is associated with the Z_2 index introduced by Kane and Mele [7] that characterizes the topology of the band structure. The skyrmion number indeed corresponds to $\Pi_2(S^2) = \mathbb{Z}$. We have made this more clear in the revised version of the manuscript.**

In contrast to a weakly coupled Bardeen-Cooper-Schrieffer-type SC, its vortices enclose a spin-1/2 degree of freedom corresponding to the fractionalized QSH order parameter.

The DQCP is characterized by an emergent NCCP¹ theory describing charge- e , spin-1/2 particles carrying $U(1)$ gauge charge, namely spinons. The QSH state corresponds to a (particle-hole) condensate of spinons, whereas the latter bind in the superconducting state. Close to the DQCP, the superconducting state in our work is very different from the generic BCS superconductor, where the core contains a Fermi liquid and in the weak-coupling regime the ground state in a vortex background is expected to be a spin singlet. In the revised version of the manuscript, we have omitted the discussion of why our realization of the superconducting state differs from BCS theory. Instead, we briefly mention this aspect in the conclusions.

3) I would also have given a reference to a famous paper by Haldane in addition to Ref 8 in: "...but proliferate slightly away from this point to generate the VBS state."

We agree and have added citations to Refs. [9, 10]. Indeed, as pointed out in Ref. [8], the field theory only holds for smooth configurations where the skyrmion size is larger than the inverse mass gap. We mention this point in the revised manuscript.

4) I found the sentence "However, it is highly non-trivial if the associated saddle point is stable." poorly written. There are many ways to decouple a quartic interaction (there are many instabilities by which distinct quadratic forms acquire an expectation value), i.e., there are competing instabilities.

In the Monte Carlo approach, the result is independent of the choice of Hubbard-Stratonovitch transformation. What we wanted to convey is that although the Hubbard-Stratonovitch field couples to the spin-orbit term, it is not obvious that a QSH state will be stabilized. For instance, one can decouple the Hubbard term in the charge sector but still obtain an antiferromagnetic ground state [11]. In the revised version, we have made this more clear.

5) I was confused by "Here, we omitted the vanishing background term ..." and "In contrast to previous analyses of Gross-Neveu criticality 29,30 we use susceptibilities rather than equal-time correlators to suppresses background contributions to the critical fluctuations." If spontaneous symmetry breaking takes place, one should replace moments by cumulants. Are the authors referring to the \mathbb{C} number arising from long-range order at the ordering wave vector?

We have changed the wording. Since $\langle \hat{O}_{r,\delta}(\tau) \rangle = 0$ by symmetry on any finite lattice, we indeed used cumulants. In general, it is better to consider susceptibilities rather than equal-time correlation functions to analyze critical points with large anomalous dimensions because the background is less dominant. A detailed discussion can be found in Ref. [12].

6) I could not appreciate the claim containing "...compare favourably ...".

We have changed **compare favourably** to **are compatible**.

7) The discussion around the sentence "One possible resolution is the scenario of 'pseudo-criticality' or 'walking coupling constant' was too short for me to be of any use.

The key idea of *pseudo-criticality* or a *walking coupling* is that the DQCP fixed point lies close to but not in the accessible manifold. For instance, it could be that a unitary CFT exists only in dimensions $2 + \epsilon + 1$ with $\epsilon > 0$, so that the RG flow becomes slow when approaching the fixed point but can never reach the latter in $2 + 1$ dimensions (where our simulations are carried out). However, the correlation length required to resolve this behaviour exceeds the achievable lattice sizes. We expanded the corresponding comment in the manuscript.

8) The discussion around the sentence "Other promising approaches that can shed further light on DQCPs make use of a lattice discretisation scheme based on projection onto a Landau level that does not break continuum symmetries." was too short for me to be of any use.

It is known that projection onto the half-filled zeroth Landau level offers the possibility of defining models of DQC with an explicit $SO(5)$ symmetry [13]. In a very recent paper, one of us and

collaborators showed that it is possible to formulate a negative sign free quantum Monte Carlo algorithm for this situation [14]. We have expanded the comment on future applications of this idea in the revised manuscript.

9) I could not follow the discussion in the paragraph “A monopole-free realisation of DQCPs is impossible in traditional settings ...”.

The basic message there is that a monopole-free realization of DQCP requires certain structures on the microscopic degrees of freedom, such as the existence of spin-1/2 particles. The discussion on anomaly was admittedly not fully self-contained for non-experts. This is because (a) this is not the main point of the paper, and (b) all the relevant formal discussions can be found in the references (in particular Ref. [4] and [15]). We have re-written the paragraph to make it more accessible.

Finally, we address the arguments underlying the hesitation to recommend publication in Nature Communications:

- **... is less creative than Ref. 22, a paper sharing many co-authors, while the methodology is the same.**

Reference 22 corresponds to [16] below and describes a direct and continuous AFM-VBS transition. Because this work only considered one of the two VBS mass terms, the transition is different from the present work and described by a field theory with a topological θ -term at $\theta = \pi$.

We did use the same, general auxiliary-field quantum Monte Carlo method, so that based on your comment, **I am not an expert in computational physics**, it is understandable that you perceive the methodological part as very similar. However, there are important differences. In Ref. [16], a model of fermions coupled to Ising spins in a transverse field was investigated, as opposed to the case of fermions coupled to a Hubbard Stratonovitch field in the present work. This has a dramatic impact on the efficiency of our updating scheme, with the present model being much better suited for large-scale simulations.

- **is too close relative to Refs. 19 and 22**

We have already commented on Ref. 22 of the original manuscript. In Ref. 19 (corresponding to [17] below), we studied an unconstrained \mathbb{Z}_2 lattice gauge theory coupled to N -flavour fermionic matter. The latter exhibits quantum phase transitions that differ significantly from the current work.

- **and adds nothing conceptually relative to Ref. 7.**

Indeed, Ref. 7, corresponding to [8], as well as discussions with Tarun Grover have greatly inspired our present work. Our results provide a lattice model for the fascinating phenomenology put forward in Ref. [8]. The fact that the field-theory arguments are borne out in numerical simulations of a lattice model is highly nontrivial and, as you mention, is demonstrated for the first time. Moreover, arguments such as yours are a slippery slope. For example, if a cold-atom experiment were to realize this type of quantum phase transition, would one refuse publication on the basis that it adds nothing new conceptually?

We hope that we have convinced you that it is—to use your own words—not at all a crime to publish this article in Nature Communications.

References

- [1] Shao, H., Guo, W. & Sandvik, A. W. Quantum criticality with two length scales. *Science* **352**, 213–216 (2016).
- [2] Nahum, A., Chalker, J. T., Serna, P., Ortuño, M. & Somoza, A. M. Deconfined Quantum Criticality, Scaling Violations, and Classical Loop Models. *Phys. Rev. X* **5**, 041048 (2015).
- [3] Nahum, A., Serna, P., Chalker, J. T., Ortuño, M. & Somoza, A. M. Emergent SO(5) Symmetry at the Néel to Valence-Bond-Solid Transition. *Phys. Rev. Lett.* **115**, 267203 (2015).
- [4] Wang, C., Nahum, A., Metlitski, M. A., Xu, C. & Senthil, T. Deconfined Quantum Critical Points: Symmetries and Dualities. *Phys. Rev. X* **7**, 031051 (2017).
- [5] Serna, P. & Nahum, A. Emergence and spontaneous breaking of approximate $O(4)$ symmetry at a weakly first-order deconfined phase transition. *arXiv:1805.03759* (2018).
- [6] Hasenbusch, M. Finite size scaling study of lattice models in the three-dimensional Ising universality class. *Phys. Rev. B* **82**, 174433 (2010).
- [7] Kane, C. L. & Mele, E. J. Z_2 Topological Order and the Quantum Spin Hall Effect. *Phys. Rev. Lett.* **95**, 146802 (2005).
- [8] Grover, T. & Senthil, T. Topological Spin Hall States, Charged Skyrmions, and Superconductivity in Two Dimensions. *Phys. Rev. Lett.* **100**, 156804 (2008).
- [9] Haldane, F. D. M. O(3) Nonlinear σ Model and the Topological Distinction between Integer- and Half-Integer-Spin Antiferromagnets in Two Dimensions. *Phys. Rev. Lett.* **61**, 1029–1032 (1988).
- [10] Read, N. & Sachdev, S. Spin-Peierls, valence-bond solid, and Néel ground states of low-dimensional quantum antiferromagnets. *Phys. Rev. B* **42**, 4568–4589 (1990).
- [11] Assaad, F. F. SU(2)-spin Invariant Auxiliary Field Quantum Monte Carlo Algorithm for Hubbard models. In Krause, E. & Jäger, W. (eds.) *High performance computing in science and engineering*, 105 (Springer, Berlin, 1998). [cond-mat/9806307].
- [12] Parisen Toldin, F., Hohenadler, M., Assaad, F. F. & Herbut, I. F. Fermionic quantum criticality in honeycomb and π -flux Hubbard models: Finite-size scaling of renormalization-group-invariant observables from quantum Monte Carlo. *Phys. Rev. B* **91**, 165108 (2015).
- [13] Lee, J. & Sachdev, S. Wess-Zumino-Witten Terms in Graphene Landau Levels. *Phys. Rev. Lett.* **114**, 226801 (2015).
- [14] Ippoliti, M., Mong, R. S. K., Assaad, F. F. & Zaletel, M. P. Half-filled Landau levels: A continuum and sign-free regularization for three-dimensional quantum critical points. *Phys. Rev. B* **98**, 235108 (2018).
- [15] Metlitski, M. A. & Thorngren, R. Intrinsic and emergent anomalies at deconfined critical points. *Physical Review B* **98**, 085140 (2018).
- [16] Sato, T., Hohenadler, M. & Assaad, F. F. Dirac Fermions with Competing Orders: Non-Landau Transition with Emergent Symmetry. *Phys. Rev. Lett.* **119**, 197203 (2017).
- [17] Assaad, F. F. & Grover, T. Simple Fermionic Model of Deconfined Phases and Phase Transitions. *Phys. Rev. X* **6**, 041049 (2016).

REVIEWERS' COMMENTS:

Reviewer #1 (Remarks to the Author):

The authors have satisfactorily addressed my comments; I can now recommend the paper for publication.